# Objective sleep assessment in >80,000 UK mid-life adults: Associations with sociodemographic characteristics, physical activity and caffeine

**Gewei Zhu[1], Michael Catt[2], Sophie Cassidy[1], Mark Birch-Machin[1], Michael Trenell[3], Hugo Hiden[4], Simon Woodman[4], Kirstie N. Anderson (ORCID)[5]***

1 Faculty of Medical Sciences, Institute of Cellular Medicine, Newcastle University, Newcastle Upon Tyne, United Kingdom, 2 National Innovation Centre for Ageing, Time Central, Newcastle Upon Tyne, United Kingdom, 3 NIHR Innovation Observatory, Gallowgate, Newcastle Upon Tyne, United Kingdom, 4 National Innovation Centre for Data, School of Computing, Newcastle University, Newcastle Upon Tyne, United Kingdom, 5 Department of Neurology, Royal Victoria Infirmary, Newcastle Upon Tyne, United Kingdom

* Kirstie.Anderson@nuth.nhs.uk

**Data Availability Statement:** The data underlying this study are held by the UK Biobank. Interested

## Abstract

### Study objectives

Normal timing and duration of sleep is vital for all physical and mental health. However, many sleep-related studies depend on self-reported sleep measurements, which have limitations. This study aims to investigate the association of physical activity and sociodemographic characteristics including age, gender, coffee intake and social status with objective sleep measurements.

### Methods

A cross-sectional analysis was carried out on 82995 participants within the UK Biobank cohort. Sociodemographic and lifestyle information were collected through touch-screen questionnaires in 2007–2010. Sleep and physical activity parameters were later measured objectively using wrist-worn accelerometers in 2013–2015 (participants were aged 43–79 years and wore watches for 7 days). Participants were divided into 5 groups based on their objective sleep duration per night (<5 hours, 5–6 hours, 6–7 hours, 7–8 hours and >8 hours). Binary logistic models were adjusted for age, gender and Townsend Deprivation Index.

### Results

Participants who slept 6–7 hours/night were the most frequent (33.5%). Females had longer objective sleep duration than males. Short objective sleep duration (<6 hours) correlated with older age, social deprivation and high coffee intake. Finally, those who slept 6–7 hours/night were most physically active.

researchers can request access to the data by contacting access@ukbiobank.ac.uk.

**Funding:** The authors received no specific funding for this work.

**Competing interests:** The authors have declared that no competing interests exist.

## Conclusions

Objectively determined short sleep duration was associated with male gender, older age, low social status and high coffee intake. An inverse 'U-shaped' relationship between sleep duration and physical activity was also established. Optimal sleep duration for health in those over 60 may therefore be shorter than younger groups.

## Introduction

Sleep is vital for the normal regulation of mood, cognition and metabolism[1]. Sleep disorders are common amongst the ageing population and sleep disturbance from any cause is increasingly recognised as a biomarker for unhealthy ageing[2]. Ideal duration of sleep remains debated and undoubtedly total sleep time changes with age, but "8 hours/night" has become defined as a somewhat idealised norm. Large cohort studies assessing self-reported sleep duration showed that 7–8 hours consistently emerged as the ideal duration for good health in 18–65 year olds[3].

Sleep architecture changes with age. During ageing, there is a degeneration of the circadian pacemaker, a progressive decline in melatonin output and decrease in rhythm amplitude which contributes to increasing sleep fragmentation and waking up earlier in the morning [4,5]. Additionally, >50% of those older than 65 years of age have chronic sleep complaints including difficulties in initiating and maintaining sleep. Sleep disturbance is associated with worse physical and mental health, cognitive impairment and falls but correlation remains debated[4]. Sleep is also influenced by gender and comorbid illnesses[6]. Females are more likely to self-report a longer sleep duration than males[7] and males tend to experience lighter sleep than females[8]. However, females self-report more sleep complaints and insomnia[9].

Chronic partial sleep deprivation has long term impact on health and longevity, including higher risks of hypertension, diabetes, obesity and depression[10]. Sleep deprivation has been proposed as a novel risk factor for dementia[11] and also insulin resistance and type 2 diabetes [12] and it is therefore important to know what the ideal sleep duration is for healthy ageing. In sleep-deprived individuals <65 years old, sleep extension by 1 hour/night for 6 weeks had significant beneficial effects on their insulin sensitivity[13]. Sleep deprivation also has impact on mental health. Those who self-reported 7–8 hours of sleep per night have higher optimism and self-esteem than those who slept <6 hours/night and >9 hours/night[14].

Caffeine is a widely consumed stimulant to counter the effect of fatigue as it can improve alertness, but it has adverse effects on the quantity and quality of sleep[15]. Caffeine intake prior to bedtime leads to reduced sleep efficiency, increased sleep onset latency, shortened second stage of sleep and reduced sleep duration in healthy adults with habitual high caffeine intake[16,17].

Sleep can be assessed subjectively by self-report such as sleep diaries or objectively using wrist-worn accelerometers and laboratory-based polysomnography[1]. There is often a mismatch between subjective and objective assessment of sleep. This is partly due to sleep state misperception which is especially common amongst people with sleep disorders[18].

Prior studies of the large UK biobank cohort have suggested that self-reported short sleep duration is a risk factor for poor cardiometabolic health[19]. Both short and long sleep durations were also found to be independently correlated with cognitive impairment[20]. The Newcastle 85+ study also found that objectively measured, disturbed sleep-wake cycles in the oldest old population was associated with multimorbidity, worse cognitive function and

reduced survival[21]. However, the prior Biobank studies have been based on self-reported sleep assessments and may have limitations. Therefore, we wished to understand the association between objective sleep assessments and age, gender, habitual caffeine intake, physical activity and social deprivation in the Biobank population cohort.

## Methods

### Population and study design

A cross-sectional analysis was conducted on baseline data and objectively assessed accelerometry data from the UK Biobank. The UK Biobank recruited approximately 500,000 participants aged 40–69 within the general population of the UK. Full-scale recruitment took place between 2007 and 2010[22]. Participants were invited to a baseline assessment visit where physical measurements and biological samples were collected. Sociodemographic, occupation, health-related and lifestyle information were collected through the use of touch-screen questionnaires which contains approximately 314 questions. To measure self-reported sleep duration, participants were asked "About how many hours sleep do you get in every 24 hours? (Please include naps)". Work patterns including shift work and night shift was asked about but sleep diary data was not collected[22].

After appropriate written consent, 103,712 participants from the UK Biobank study were later invited to wear wrist acceleration sensors (Axivity AX3 triaxial accelerometer) on their dominant wrist continuously for 7 consecutive days. Recruitment occurred between 2013 and 2015 (participants aged between 43 and 79). This allowed objective measurements of their physical activity and sleep-wake patterns[22,23].

### Sleep categories

Accelerometry data obtained from the UK Biobank were processed using R Package GGIR version 1.7–1. This has been validated and published as open source[24]. The generic algorithm has been assessed in a large UK study[25]. The accuracy of the algorithm at detecting sleep period time window has also been tested[26]. Only participants with at least 5 days of complete accelerometry data were included in the current study. Both sleep duration and sleep efficiency were examined. Participants were categorised into 5 groups based on their sleep durations using previously published self-report thresholds[27]. (1) Subjects who slept <5 hours/night. (2) Subjects who slept 5–6 hours/night. (3) Subjects who slept 6–7 hours/night. (4) Subjects who slept 7–8 hours/night. (5) Subjects who slept >8 hours/night. Group 1 and 2 are used to investigate the impact of extremely short objective sleep durations.

### Baseline measurements

Sociodemographic characteristics (including age, gender and deprivation scores) and dietary information such as tea and coffee intakes were collected from the touch-screen questionnaires at recruitment.

The age of each participant at accelerometry data collection was calculated. Participants were categorised into 4 age groups (43–49 years, 50–59 years, 60–69 years and 70–79 years). Gender was recorded at recruitment. Townsend deprivation index was calculated immediately prior to participant joining the UK Biobank based on the national census data. Participants were each assigned a score depending on the location of their post codes at recruitment[28]. This index takes account of home ownership, car ownership and employment status. Townsend deprivation index was divided into quintiles (0 represents the least deprived individuals and 4 represents the most deprived individuals).

At the baseline assessment visit, participants were also asked about how many cups of tea and coffee (include decaffeinated coffee) they drink per day over the last year. If participants answered >20 for tea or >10 for coffee, they were asked to confirm their answers. If they are unsure, participants could provide an estimate or select 'Do not know'. For those participants who indicated they drink <1 or ≥1 cup coffee per day, they were then asked 'What type of coffee do you usually drink?"[28].

## Physical activity measurements

Acceleration levels (measured in milli-g (mg)) of each participants were extracted from the accelerometry data which is a measurement of physical activity level[23]. Acceleration measurements were separated into each wear day allowing comparison between week days and weekend sleep and activity pattern. This method of assessing physical activity has previously been validated and published in detail using the biobank cohort which has also compared to other accelerometry devices[23].

## Statistical analysis

Both Biobank baseline data and accelerometry data were analysed using IBM SPSS Statistics version 24 (Armonk, New York, USA). Individuals with missing data on accelerometry measurements were excluded. Chi-square test was used to investigate the association between sleep groups and categorical variables. Once a significant difference is detected between any sleep groups, z-test was used to compare column proportions of each variables and p-values were adjusted using the Bonferroni method. If columns in the same row have been assigned the same letter, then their column proportions do not differ from each other significantly. Monte carlo method with 99% confidence level was used to estimate the exact significant level. Kruskal-Wallis H test was used to investigate the association between sleep groups and sleep efficiency with continuous variables.

Binary logistic regression was used to investigate the odds of being male, aged over 70 years, live in the most deprived area and reporting high coffee intake across the 5 sleep groups. Adjusted odds ratios (OR), with 95% CIs were reported. Logistic regression models were adjusted for: age (reference = '43–49'); gender (reference = 'Female') and Townsend deprivation index (reference = 'Least deprived). Significance for all statistical tests was set at p<0.05)

# Results

## Sociodemographic characteristics

Of the total UK Biobank participants, after excluding those with missing data, there were 82,995 participants in total. 14,333 (17.3%) slept <5 hours/night, 21,559 (26.0%) slept 5–6 hours/night, 27,783 (33.5%) slept 6–7 hours/night, 15,503 (18.7%) slept 7–8 hours/night and 3,817 (4.6%) slept >8 hours/night (Fig 1). There was no statistically significant difference in sleep duration between weekdays and weekends across the entire group or within those under and over the age of 65.

The percentage of males that slept <5 hours/night was significantly higher, while more females slept >7 hours/night (Fig 1). The greatest proportion of participants were within the '60–69 years' age group. According to the Townsend deprivation index, socioeconomic status increased across the sleep groups up to the second quintile then it decreased across the groups. Overall, no significant differences in terms of gender, age and Townsend deprivation index were detected between '7–8 hours' and '>8 hours' groups (Table 1).

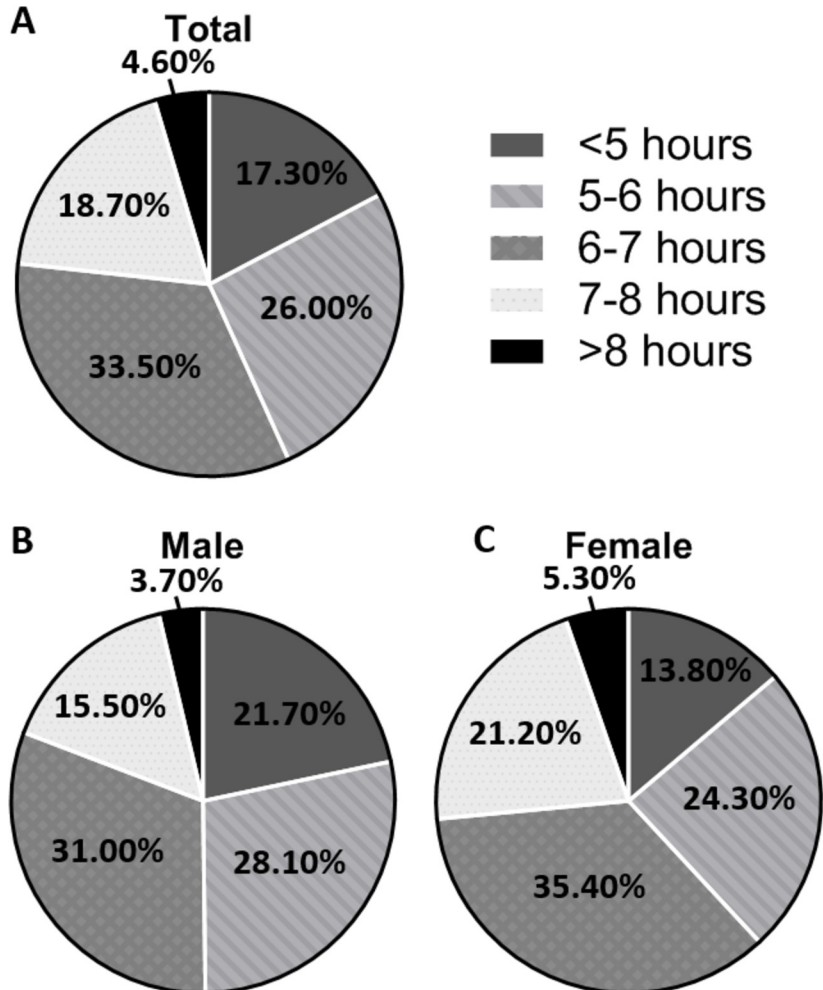

**Fig 1. A: Percentage of total participants in each sleep group (n = 82995). B: Percentage of males in each sleep group (n = 36293). C: Percentage of females in each sleep group (n = 46702).** Those who sleep 6–7 hours/night are more prevalent and females have longer objective sleep duration than males.

Those with the shortest sleep duration (<5 hours/night) were 115% (OR (95% CI) 2.15 (2.06 to 2.26)), 15% (OR (95% CI) 1.15 (1.08 to 1.21)) and 221% (OR (95% CI) 3.21 (2.24 to 4.61)) more likely to be male, aged over 70 years, and live in the most deprived area, respectively, compared with the '7–8 hours' sleep group (Table 2).

Sleep efficiency was significantly higher in females, those aged 60–69 years old and participants with higher social status (p<0.001) (Table 3).

### Habitual coffee intake

The percentage of participants with prior high coffee intake (>4 cups/day) was significantly higher in those who slept <5 hours/night (Table 1). Those with the shortest sleep duration (<5 hours/night) were 26% (OR (95% CI) 1.26 (1.17 to 1.36)) more likely to be have high coffee intake (>4 cups/day) compared with the '7–8 hours' sleep group (Table 2). When investigating the relationship between coffee type and sleep duration; the percentage of participants who drink decaffeinated coffee is significantly higher in those who sleep >7 hours/night

**Table 1. Sociodemographic and dietary characteristics of the 5 sleep groups (n = 82995).**

| | Percentage within each sleep group | | | | |
|---|---|---|---|---|---|
| | <5 hours (n = 14333) | 5–6 hours (n = 21559) | 6–7 hours (n = 27783) | 7–8 hours (n = 15503) | >8 hours (n = 3817) |
| **Gender** | | | | | |
| Male (%) | 54.9 a | 47.3 b | 40.5 c | 36.3 d | 35.4 d |
| Female (%) | 45.1 a | 52.7 b | 59.5 c | 63.7 d | 64.6 d |
| **Age groups (years)** | | | | | |
| 43–49 (%) | 6.0 a | 7.7 b | 7.5 b | 7.1 b | 6.2 a,b |
| 50–59 (%) | 27.0 a | 30.0 b | 28.7 b,c | 25.9 a | 26.0 a,c |
| 60–69 (%) | 42.8 a,b | 42.2 b | 44.2 a,c | 45.9 c | 46.8 c |
| 70–79 (%) | 24.2 a | 20.1 b,c | 19.6 c | 21.1 b | 21.0 b,c |
| **Townsend deprivation index quintile** | | | | | |
| 0 (Least deprived) (%) | 20.5 a | 22.0 b | 23.9 c | 24.3 c | 24.4 c |
| 1 (%) | 20.2 a | 20.9 a | 22.0 b | 22.6 b | 23.0 b |
| 2 (%) | 19.9 a | 20.6 a,b | 20.7 a,b | 21.5 b | 19.9 a,b |
| 3 (%) | 20.6 a | 20.3 a | 19.1 b | 18.7 b | 18.8 a,b |
| 4 (Most deprived) (%) | 18.7 a | 16.2 b | 14.2 c | 13.0 d | 13.9 c, d |
| **Average coffee intake per day** | | | | | |
| 0/<1 cup (%) | 28.3 a | 26.8 b | 27.3 a,b | 28.1 a | 28.8 a,b |
| 1–3 cups (%) | 50.3 a | 53.6 b | 54.1 b | 54.7 b | 53.6 b |
| >4 cup (%) | 21.3 a | 19.6 b | 18.6 b,c | 17.2 d | 17.5 c,d |
| **Coffee type** | | | | | |
| Decaffeinated coffee (any type) (%) | 17.6 a | 18.1 a,b | 19.1 b | 20.5 c | 21.6 c |
| Instant coffee (%) | 52.8 a | 51.5 a,b | 50.6 b,c | 49.7 c | 51.9 a,b,c |
| Ground coffee (include espresso, filtered etc) (%) | 27.7 a | 28.8 a | 28.9 a | 28.3 a | 24.6 b |
| Other types of coffee (%) | 1.6 a | 1.3 a,b | 1.2 b | 1.3 a,b | 1.6 a,b |
| Prefer not to answer (%) | 0.1 a | 0.0 a | 0.1 a | 0.1 a | 0.1 a |

Column proportions test was carried out and column proportions (for each row) are compared using a z-test. Each letter denotes a subset of sleep group categories whose column proportion do not differ significantly from each other at 0.05 level. Short sleep duration is significantly associated with male gender, older age, social deprivation and high coffee intake.

(Table 1). The type of caffeinated coffee (instant versus espresso) did not affect sleep duration. In addition, males drank significantly more coffee than females (Table 4).

Sleep efficiency was negatively correlated with coffee intake (p = 0.001 and p<0.001, respectively). Not only did the amount of coffee drank affect the sleep efficiency, but those who drank decaffeinated coffee had significantly higher sleep efficiency (p = 0.023) (Table 3).

**Table 2. OR (95% CI) of being male, aged over 70 years, live in the most deprived area and reporting high coffee intake across sleep groups.**

| | Male | Aged >70 years | Social deprivation | >4 cups of coffee/day |
|---|---|---|---|---|
| **7–8 hours** | 1.00 | 1.00 | 1.00 | 1.00 |
| **<5 hours** | 2.15 (2.06 to 2.26) | 1.15 (1.08 to 1.21) | 3.21 (2.24 to 4.61) | 1.26 (1.17 to 1.36) |
| **5–6 hours** | 1.62 (1.55 to 1.68) | 0.93 (0.88 to 0.98) | 1.58 (1.09 to 2.30) | 1.13 (1.06 to 1.21) |
| **6–7 hours** | 1.21 (1.17 to 1.26) | 0.90 (0.86 to 0.95) | 1.29 (0.89 to 1.87) | 1.07 (1.00 to 1.14) |
| **>8 hours** | 0.97 (0.90 to 1.04) | 1.03 (0.95 to 1.12) | 1.55 (0.86 to 2.81) | 0.98 (0.87 to 1.10) |

Statistical models were adjusted for age, gender and Townsend Deprivation Index.

**Table 3. Association between sociodemographic characteristics, coffee intake, social deprivation and sleep efficiency.**

| | Sleep efficiency (mean + standard deviation) | P-value |
|---|:---:|:---:|
| **Gender** | | <0.001 |
| Male | 0.811 ± 0.117 | |
| Female | 0.829 ± 0.110 | |
| **Age (years)** | | <0.001 |
| 43–49 | 0.821 ± 0.109 | |
| 50–59 | 0.820 ± 0.113 | |
| 60–69 | 0.823 ± 0.113 | |
| 70–79 | 0.820 ± 0.118 | |
| **Townsend deprivation index quintile** | | <0.001 |
| 0 (Least deprived) | 0.824 ± 0.114 | |
| 1 | 0.824 ± 0.113 | |
| 2 | 0.822 ± 0.114 | |
| 3 | 0.820 ± 0.112 | |
| 4 (Most deprived) | 0.816 ± 0.116 | |
| **Average coffee intake per day** | | <0.001 |
| 0/<1 cup | 0.821 ± 0.113 | |
| 1–3 cups | 0.823 ± 0.115 | |
| >4 cups | 0.818 ± 0.117 | |
| **Coffee type** | | <0.001 |
| Decaffeinated coffee (any type) | 0.825 ± 0.115 | |
| Instant coffee | 0.821 ± 0.115 | |
| Ground coffee (include espresso, filtered etc) | 0.821 ± 0.117 | |
| Other types of coffee | 0.816 ± 0.114 | |

Higher sleep efficiency correlated with lower coffee intake and decaffeinated coffee type. The differences between coffee intake and between different coffee types were all found to be statistically significant (p<0.05).

## Association between acceleration level and sleep duration

Physical activity levels were monitored continuously throughout the day. This allowed the determination of the most active and least active 5 hours of each day using the detected acceleration level measured in milli-g (Table 5). An inverse 'U-shaped' association between activity

**Table 4. Association between gender and average coffee intake per day.**

| | Percentage within gender | |
|---|:---:|:---:|
| | **Female** | **Male** |
| **Average coffee intake per day** | | |
| 0/<1 cup (%) | 29.7 a | 24.7 b |
| 1–3 cups (%) | 53.4 a | 53.4 a |
| >4 cup (%) | 16.8 a | 21.8 b |
| Do not know (%) | 0.1 a | 0.1 a |
| Prefer not to answer (%) | 0.0 a | 0.0 a |

Column proportions test was carried out and column proportions (for each row) are compared using a z-test. Each letter denotes a subset of sleep group categories whose column proportion do not differ significantly from each other at 0.05 level. Males were found to consume significantly more cups of coffee per day compared to females.

**Table 5. Association between activity level and sleep duration (p<0.001).**

|  | <5 hours | 5–6 hours | 6–7 hours | 7–8 hours | >8 hours |
|---|---|---|---|---|---|
| Average acceleration during M5 (mean ± SD) (mg) | 55.56 ± 22.76 | 58.21 ± 20.94 | 58.39± 20.70 | 57.39 ± 19.70 | 54.23 ± 18.98 |
| Average acceleration during L5 (mean ± SD) (mg) | 4.58 ± 3.11 | 3.76 ± 1.86 | 3.44 ± 1.94 | 3.27 ± 1.48 | 3.24 ± 2.11 |
| ΔM5L5 (mean + SD) (mg) | 50.98 ± 22.32 | 54.45 ± 20.87 | 54.95± 20.59 | 54.11 ± 19.63 | 50.99 ± 18.44 |

Acceleration levels were monitored throughout the day which allowed the determination of average acceleration during the most active 5 hours of the day (M5) and the least active 5 hours of the same day (L5). The difference between these 2 measurements (ΔM5L5). Those who slept 6–7 hours/night were found to be more active compared to other sleep groups.

level and sleep duration was observed. Average acceleration level during the most active 5 hours of the day (M5) was found to be the highest in those who slept 6–7 hours and the lowest in participants who slept >8 hours/night (Fig 2A). On the other hand, the average acceleration level during the least active 5 hours of the day (L5) decreased across the sleep groups. It was the highest in those who slept <5 hours/night, while the difference between the other 4 sleep groups was relatively small. As a biomarker for healthy ageing, the difference between the most and least active 5 hours of the same day (ΔM5L5) was calculated from the M5 and L5 readings[21] and it was the highest in '6–7 hours' sleep group (Fig 2B). It was significantly lower in those who slept <5 hours/night and >8 hours/night (p<0.001).

## Discussion

Using the largest accelerometer cohort to date, this study found that short objective sleep duration is associated with male gender, older age, social deprivation and habitual high caffeine intake. An inverse 'U-shaped' relationship between objective sleep duration and physical activity level was also identified suggesting that the most physically active slept between 6–7 hours. All prior UK biobank sleep studies have used the self-reported sleep data. An association between worse cardiometabolic health and impaired task performance with sleep duration <7 hours or >9 hours per night was reported[20,29]. Previous large cohort studies also utilised wrist-worn actigraphy to assess sleep objectively[30–34]. There have been smaller accelerometry studies within the Whitehall cohort (n = 3749) showing an association between longer duration and higher intensity of daily moderate-to-vigorous physical activity level and successful ageing[35]. However, no one has to date assessed the objective sleep-wake patterns and physical activity using accelerometry in the Biobank cohort.

We found 43.3% of the UK Biobank cohort had an objective sleep duration of <6 hours/ night. The American Academy of Sleep Medicine and Sleep Research Society have recommended >7 hours of sleep per night for 18–60 years old adults for good health and reduced mortality[36]. Only 23.3% of the UK Biobank participants reached this recommended sleep duration, this may mean that in an older population the ideal sleep duration may not be 7–8 hours but instead somewhat shorter when assessed using this accelerometer based objective method rather than self-report. Age adjusted norms remain debated and are key when considering recommendations and advice for an ageing population. In fact, careful assessment of healthy older adults excluding sleep disorders has shown a clear decrease in sleep need with age without effects on daytime alertness[37]. Social jet lag (sleep duration significantly longer at weekends compared to the working week) was also assessed in the UK Biobank cohort as a possible marker of societal sleep restriction and therefore one possible reason for objective short sleep time. Participants' sleep duration was not significantly longer on Fridays or Saturdays compared to other days of the week. This could be due to many of the participants having reached retirement age and therefore, their sleep duration is not significantly affected by

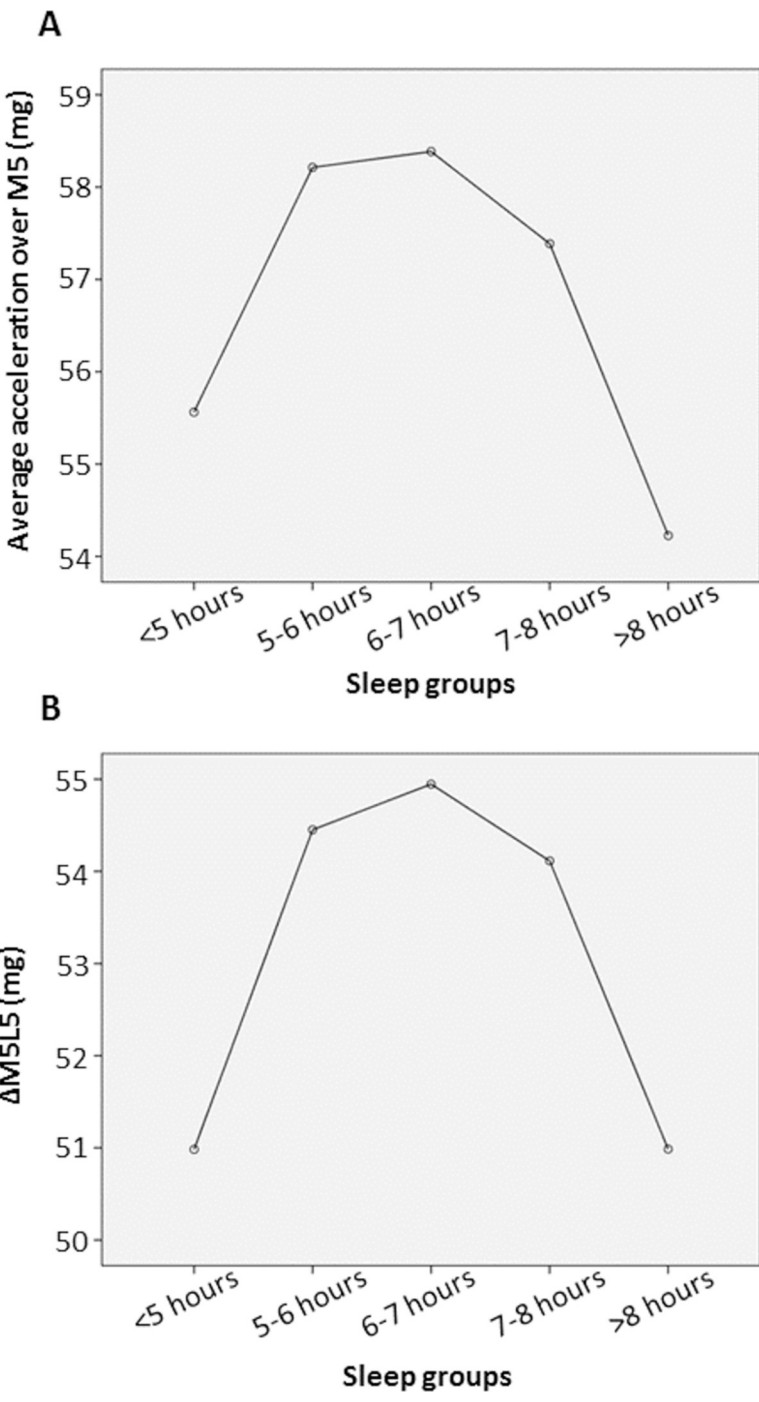

**Fig 2. Distribution of acceleration measured in milli-g (mg) across the 5 sleep groups (p<0.001).** A: average acceleration over the most active 5 hours of the day (M5). B: Difference in acceleration between the most active 5 hours (M5) and least active 5 hours (L5) of the same day (ΔM5L5). Those who sleep 6–7 hours/night had the highest acceleration level.

longer working hours during the week. However, it could also suggest that their shorter sleep times were not indicative of restricted or inadequate sleep and supports 6–7 hours being the likely optimum sleep duration for many over the age of 60.

When comparing physical activity levels of our 5 sleep groups; those who slept 6–7 hours/night were most active with an inverse U-shaped curve and decreased activity in very short and long sleepers. Apart from healthy sleeping habits, the beneficial effects of an active lifestyle on health are well known. The ability of objectively measured physical activity in improving sleep has been shown. Those who have met the recommended World Health Organisation physical activity guidelines (>150mins of moderate intensity or >75mins vigorous intensity per week) self-reported shorter sleep latency, less early awakenings and less leg cramps during sleep fewer usage of sleeping pills[38]. Our findings may suggest that physical activity can promote healthy sleep duration, but it may also be that healthy sleeping habits promote a more active lifestyle. Strong evidence suggests inadequate moderate-to-vigorous physical activity is closely associated with increased risks of metabolic diseases and all-cause mortality after controlling for age, gender, race and weight[39]. Physical inactivity also leads to direct and indirect costs which result in considerable financial burdens on the society[40]. Due to the cross-sectional nature of the study, we are unable to prove the direction of relationship between physical activity and sleep.

A significant 'U-shaped' association between sleep duration and metabolic diseases has previously been published with short sleep associating with worse cardiometabolic health. However, after adjusting for confounders, this association only remained in short sleepers (<6 hours/night)[3]. In this study, shorter objective sleep duration and lower sleep efficiency were found to correlate with the male gender, social deprivation and high coffee intakes. These are groups previously shown to be at higher risk of poor cardiometabolic health and increased mortality. Our results are also consistent with previous studies showing the correlation of lower socioeconomic status with longer sleep latency and poorer quality of sleep as measured by actigraphy and polysomnography[41,42]. Many factors including poor mental health and physical diseases can affect sleep quality and they are more prevalent in socially deprived individuals. This may well confound any correlations between social deprivations and sleep parameters[41,42]. Within our cohort, a habitual high coffee intake (>4 cups/day) was associated with longer sleep latency, more awakenings during the night and sleep complaints[43]. The difference in sleep efficiency between different types of coffee is relatively small but there was a significant decrease in sleep time in those with habitual high coffee intake and this group were more likely to be male. Our results demonstrated that caffeinated coffee does have an impact on sleep efficiency compared to decaffeinated coffee. Although 3–4 cups of coffee per day was not significantly associated with health risks. The prevalence of coffee drinking and usual intake increases with age[44] and coffee is the main source of dietary caffeine in the western society so may be one factor in short sleep in an older population. Therefore, the relationship between habitual high coffee intake and sleep in ageing populations is worthy of further investigation[45]. However, limitations of the study were that habitual tea/coffee intake data were collected some years prior to accelerometry data collection. This means that any association detected needs to be interpreted with care.

The current study found a discrepancy between self-reported and accelerometry assessments of sleep. The prior self-reported sleep duration of the Biobank cohort showed that 78% of participants had self-reported a sleep duration >7 hours/night, but accelerometry data showed that only 23% of participants slept >7 hours/night. Therefore, participants may overestimate their sleep duration. Moreover, we found that females self-reported shorter sleep durations but they have significantly longer objective sleep duration and higher sleep efficiency than males. Similar results were also found in the Rotterdam study[46]. Within the prior UK Biobank studies, females were more likely to self-report insomnia symptoms and shorter sleep duration[20]. However, evidence from polysomnographic measures and quantitative electro-encephalographic analysis does not support this[47]. Our results suggested that sleep state

misperception may be more common amongst females. However, any potential association needs to be interpreted with care as there is on average a 5 year time lag between self-report sleep duration and accelerometry assessment of sleep. The sleep pattern of some of the participants might have changed during this period of time.

The main strength of the current UK Biobank study is the large sample size and extensive information collected. It provided detailed, objective measurements of sleep duration and efficiency, physical activity, sociodemographic and dietary characteristics. The specific accelerometers allowed the possibility of measuring physical activity level and sleep/wake patterns in a single device using open access algorithms. This is more likely to be an accurate measure of sleep duration compared to limited, short questions of self report sleep duration previously collected in this cohort. This is by far the largest UK accelerometry cohort and subsequent long term follow up studies will be able to determine whether an optimum sleep duration and physical activity level predicts healthy ageing. However, we accept the limitations of a cross-sectional study design and the time lag between accelerometry data and other biometric assessments within the Biobank cohort. Many other potential confounders such as physical or psychiatric disease were not assessed due to the time lag between accelerometry data collection and original biometric variables and we accept that future studies would ideally include these variables. Finally, due to the constraints of the Biobank, other sources of caffeine intake such as coke and energy drink consumption were not taken into account.

In conclusion, our study provided further insights into the relationship between objective sleep duration, sociodemographic and physical activity using the largest ever accelerometer cohort. Males, socially deprived individuals and habitual high coffee drinkers were found to have shorter objective sleep duration and lower sleep efficiency. 6–7 hours of sleep per night was associated with the highest physical activity levels. This raises the possibility that objectively assessed 6–7 hours of sleep per night may be optimal for health, at least for those aged over 60 years old. Understanding the association with sleep and health could help to design and optimise interventions to targeted groups and therefore reduce the adverse health impact of poor sleep.

## Acknowledgments

The authors would like to thank the UK Biobank participants for agreeing to volunteer in this research and investigators for making this study possible.

## Author Contributions

**Conceptualization:** Michael Catt, Sophie Cassidy, Mark Birch-Machin, Michael Trenell, Kirstie N. Anderson.

**Data curation:** Gewei Zhu, Sophie Cassidy, Simon Woodman.

**Formal analysis:** Gewei Zhu, Sophie Cassidy, Hugo Hiden, Simon Woodman, Kirstie N. Anderson.

**Methodology:** Hugo Hiden, Simon Woodman, Kirstie N. Anderson.

**Software:** Gewei Zhu, Michael Catt, Hugo Hiden.

**Supervision:** Mark Birch-Machin, Michael Trenell, Kirstie N. Anderson.

**Writing – original draft:** Gewei Zhu, Kirstie N. Anderson.

**Writing – review & editing:** Michael Catt, Sophie Cassidy, Mark Birch-Machin, Michael Trenell.

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
