## [Decision Letter · Decision Letter 0]

26 Sep 2019

PONE-D-19-20962

Objective sleep assessment in >80,000 UK mid-life adults: associations with sociodemographic characteristics, physical activity and caffeine

PLOS ONE

Dear  Dr. Anderson,

Thank you for submitting your manuscript to PLOS ONE. After careful consideration, we feel that it has merit but does not fully meet PLOS ONE’s publication criteria as it currently stands. Therefore, we invite you to submit a revised version of the manuscript that addresses the points raised during the review process.

The two reviewers and this editor have identified several strengths of this manuscripts. However this manuscript  also exists several weaknesses, especially in statistical analysis. All the comments raised by the two reviewers and this editor should be addressed in the reversion. 

We would appreciate receiving your revised manuscript by Nov 10 2019 11:59PM. To enhance the reproducibility of your results, we recommend that if applicable you deposit your laboratory protocols in protocols.io, where a protocol can be assigned its own identifier (DOI) such that it can be cited independently in the future. For instructions see: http://journals.plos.org/plosone/s/submission-guidelines#loc-laboratory-protocols

We look forward to receiving your revised manuscript.

Kind regards,

yinglin xia, Ph.D.

Academic Editor

PLOS ONE

Journal Requirements:

Additional Editor Comments:

This study has the strengths including 1) large data set and 2) using objective measure of sleep.

However, this study also has the weaknesses. The most weakness is the statistical analysis.

1. We do not know what exact methods or models were used for this study.

2. z-test is not clear

3. Actually more details of statistical analysis are needed.

For example how the continuous variable and categorical variables were analyzed.

What tests were used, such as t-test or ANOVA or non-parametric t-test and ANOVA were used?

Did the study tested the data distributions?

For categorical variables, what tests were used, Chi-square test or Fisher exact test? Did the study use contrast to compare the different levels of categories?

4. Since in the data sets there have several demographics variable such as age and gender, and clinical variables. A statistical model is more appropriate for the analysis instead of simple tests.

5. How many percentage of missing data? Excluding those missing data from the analysis is a limitation. The appropriate approach should include the missing data and using a appropriate method or model to deal with the missing data.

Reviewers' comments:

Reviewer's Responses to Questions

**Comments to the Author**

1. Is the manuscript technically sound, and do the data support the conclusions?

Reviewer #1: Yes

Reviewer #2: Yes

2. Has the statistical analysis been performed appropriately and rigorously? 

Reviewer #1: I Don't Know

Reviewer #2: Yes

3. Have the authors made all data underlying the findings in their manuscript fully available?

Reviewer #1: Yes

Reviewer #2: Yes

4. Is the manuscript presented in an intelligible fashion and written in standard English?

Reviewer #1: Yes

Reviewer #2: Yes

5. Review Comments to the Author

Reviewer #1: In this study, objective sleep efficiency and duration were investigated by Zhu et al using wrist-worn devices in large cohort of aging 43-79 in UK. It was concluded that shorter sleep duration was associated with male gender, older age, social deprivation and higher caffeine intake. More importantly, the authors found that 6-7 hours sleep/night was associated with the highest physical activity level, and discussed the differences between objective assessment and subjective questionnaire.

Minors:

It is impossible to include participants aged <43 for the current study.

Letters denoting column proportion z-test in note of Table 1 are confusing, better if authors can explain them more.

Line 174-320: duplicate reference

Line 166-174: font and size need to edit

Line 84: “the oldest old”?

Line 56: “difficulties” in?

Line 93: age “40-69”; line 31: aged “43-69”?

Line 166-168: compared to what? Refer to Fig 1 or Table 1?

Line 330: gender p-value: p < 0.001 in table 2.

Reviewer #2: The authors assessed a relationship between sociodemographics and sleep duration determined by objective actigraphy measures in a large population based sample and showed relevant factors to sleep duration. Also the authors the demonstrated inverse U-shaped relationship between sleep duration and physical activity by objective sleep measures.

Introduction and methods section are clear and well written.

Page 9-16, lines 166-320,

Large font texts and references list are incorrectly inserted here. Please check and correct them.

Are there any correlations between sleep time, measured by objective measures, and subjective sleepiness or sleep complaints?

Can actigraphy differentiate sleep and immobile waking state?

Information about smoking and alcohol intake can interfere with sleep and should be presented if possible.

Statistical analysis should be described in more detail, as this study address a large sample.

6. PLOS authors have the option to publish the peer review history of their article (what does this mean?). If published, this will include your full peer review and any attached files.

Reviewer #1: Yes: Guo Luo

Reviewer #2: No

---

## [Author Response · Author response to Decision Letter 0]

14 Oct 2019

Objective sleep assessment in >80,000 UK mid-life adults: associations with sociodemographic characteristics, physical activity and caffeine. PONE-D-19-20962

Dear Dr Yinglin Xia,

We would like thank the reviewer for their time and helpful comments. Amendments to the original text can be identified using track.

Please find our responses to the comments as follows.

Editor comments:

1. No statistical model was previously carried out. Additional explanation on the statistical tests used are added in the method section.

2. More explanation on the z-test are added – z-test was used instead of standard t-test to determine whether 2 population mean was different because of the large sample size in the current study.

3. Details of the statistically analysis are now added as requested – Monte carlo method with 99% confidence level was used to estimate the exact significant level. ANOVA was used on continuous variables and Chi-square test was used on categorical variables. Kolmogorov-smirnov test indicates that sleep duration does not follow a normal distribution (D(84411)=0.065, p<0.001).

4. Statistical model results are added as suggested.

5. 11.6% (n=10918) were excluded from the current study including 1.6% (n=1425) due to participant drop-outs over-time. Comparison between the population in the current study and those excluded was undertaken, there are no significant differences in terms of age and gender, and therefore we believe that there is unlikely to be a selection bias.

Reviewer 1 comments:

Letters denoting column proportion z-test in note of Table 1 are confusing, better if authors can explain them more.

Additional explanation is added in the method section as suggested.

Line 174-320: duplicate reference

This might be an error in printing from Microsoft word. When viewing on the computer or print from PDF, this problem does not occur.

Line 166-174: font and size need to edit

This might be an error in printing from Microsoft word. When viewing on the computer or print from PDF, this problem does not occur.

Line 84: “the oldest old”?

This refers to the 85+ population.

Line 56: “Difficulties” in?

This refers to difficulties in initiating and maintaining sleep. The sentence has been amended to make it clearer.

Line 93: age “40-69”; line 31: aged “43-69”?

There is a time lag between self-reported data and accelerometry data. Participants were aged 40-69 years during in baseline assessment around 2007-2010, but their age are between 43-79 years when accelerometry data were taken around 2013-2015. 

Line 166-168: compared to what? Refer to Fig 1 or Table?

This sentence has now been rearranged to improve clarity.

Line 330: gender p-value: p<0.001 in table 2

This typo has now been changed. 

Reviewer 2 comments:

Page 9-16, lines 166-320, large font texts and references list are incorrectly inserted here. Please check and correct them.

This might be an error in printing from Microsoft word. When viewing on the computer or print from PDF, this problem does not occur.

Are there any correlations between sleep time, measured by objective measures, and subjective sleepiness or sleep complaints?

A previous study by Chaput JP et al. (2013) found a ‘U-shaped’ association between subjective sleep duration and metabolic measures and the current study found a similar association between that and objective sleep duration. 

Can actigraphy differentiate sleep and immobile waking state?

Vincent T. van Hees et al. (2015) found that GGIR over-estimates sleep duration by 31 minutes with 83% accuracy. If arm angle does not change greater than 5 degrees in 5 minutes then it is considered a bout of sleep. Therefore actigraphy remains an acceptable tool to estimate the major sleep period, widely used across a number of different population cohort studies.

Information about smoking and alcohol intake can interfere with sleep and should be presented if possible.

Smoking status and alcohol intake information are available, but due to the ~5 years’ time lag between these data and accelerometry data, smoking and alcohol data is not presented in this paper.

Statistical analysis should be described in more details, as this study address a large sample.

More information on the statistical tests are added in the method section as suggested.

Yours sincerely

Miss Gewei Zhu and Dr Kirstie Anderson

---

## [Decision Letter · Decision Letter 1]

29 Oct 2019

PONE-D-19-20962R1

Objective sleep assessment in >80,000 UK mid-life adults: associations with sociodemographic characteristics, physical activity and caffeine

PLOS ONE

Dear Miss Gewei Zhu and Dr Kirstie Anderson,

Thank you for submitting your manuscript to PLOS ONE. After careful consideration, we feel that it has merit but does not fully meet PLOS ONE’s publication criteria as it currently stands. Therefore, we invite you to submit a revised version of the manuscript that addresses the points raised during the review process.

Since the Kolmogorov-smirnov test indicates that sleep duration does not follow a

normal distribution (D(84411)=0.065, p<0.001), the non-parametric **Kruskal-Wallis ANOVA should be used.The study must re-analyze this part. **

We would appreciate receiving your revised manuscript by Dec 13 2019 11:59PM. To enhance the reproducibility of your results, we recommend that if applicable you deposit your laboratory protocols in protocols.io, where a protocol can be assigned its own identifier (DOI) such that it can be cited independently in the future. For instructions see: http://journals.plos.org/plosone/s/submission-guidelines#loc-laboratory-protocols

We look forward to receiving your revised manuscript.

Kind regards,

yinglin xia, Ph.D.

Academic Editor

PLOS ONE

Reviewers' comments:

Reviewer's Responses to Questions

**Comments to the Author**

1. If the authors have adequately addressed your comments raised in a previous round of review and you feel that this manuscript is now acceptable for publication, you may indicate that here to bypass the “Comments to the Author” section, enter your conflict of interest statement in the “Confidential to Editor” section, and submit your "Accept" recommendation.

Reviewer #1: (No Response)

Reviewer #2: All comments have been addressed

2. Is the manuscript technically sound, and do the data support the conclusions?

Reviewer #1: (No Response)

Reviewer #2: Yes

3. Has the statistical analysis been performed appropriately and rigorously? 

Reviewer #1: (No Response)

Reviewer #2: N/A

4. Have the authors made all data underlying the findings in their manuscript fully available?

Reviewer #1: (No Response)

Reviewer #2: Yes

5. Is the manuscript presented in an intelligible fashion and written in standard English?

Reviewer #1: (No Response)

Reviewer #2: Yes

6. Review Comments to the Author

Reviewer #1: (No Response)

Reviewer #2: The authors have well addressed my concerns.

Page 9-16, lines 166-320, large font texts and references list are incorrectly inserted here. Please check and correct them.

However, this has not been revised. Can the authors check PDF version of the manuscript before submission?

7. PLOS authors have the option to publish the peer review history of their article (what does this mean?). If published, this will include your full peer review and any attached files.

Reviewer #1: No

Reviewer #2: Yes: Keisuke Suzuki

---

## [Author Response · Author response to Decision Letter 1]

19 Nov 2019

Dear Yinglin Xia,

We would like thank the reviewer for their time and helpful comments. Amendments to the original text can be identified using track.

Please find our responses to the comments as follows.

Comment: Since the Kolmogorov-smirnov test indicates that sleep duration does not follow a normal distribution (D(84411)=0.065, p<0.001), the non-parametric Kruskal-Wallis ANOVA should be used. The study must re-analyze this part. 

Response: Analysis between continuous variables has been repeated using the Kruskal-Wallis test. Method and results section has been amended. Largely results were unchanged but all amendments are clearly marked.

Reviewer comments to the author

Reviewer 2: Page 9-16, lines 166-320, large font texts and references list are incorrectly inserted here. Please check and correct them.

However, this has not been revised. Can the authors check PDF version of the manuscript before submission?

Response: No error in font size and reference list are detected on either of our computers, therefore we are unsure whether this is due to errors when uploading online.

Yours sincerely

Miss Gewei Zhu and Dr Kirstie Anderson

---

## [Editor Report · Decision Letter 2]

22 Nov 2019

Objective sleep assessment in >80,000 UK mid-life adults: associations with sociodemographic characteristics, physical activity and caffeine

PONE-D-19-20962R2

Dear Dr. Anderson,

We are pleased to inform you that your manuscript has been judged scientifically suitable for publication and will be formally accepted for publication once it complies with all outstanding technical requirements.

With kind regards,

yinglin xia, Ph.D.

Academic Editor

PLOS ONE
---

## [Editor Report · Acceptance letter]

18 Dec 2019

PONE-D-19-20962R2 

Objective sleep assessment in >80,000 UK mid-life adults: associations with sociodemographic characteristics, physical activity and caffeine 

Dear Dr. Anderson:

I am pleased to inform you that your manuscript has been deemed suitable for publication in PLOS ONE. Congratulations! Your manuscript is now with our production department. 

With kind regards,

on behalf of

Dr. yinglin xia 

Academic Editor

PLOS ONE